# Toll-like Receptor Type 2 and 13 Gene Expression and Immune Cell Profiles in Diploid and Triploid Sterlets (*Acipenser ruthenus*): Insights into Immune Competence in Polyploid Fish

**DOI:** 10.3390/ijms26093986

**Published:** 2025-04-23

**Authors:** Olga Jablonska, Sara Duda, Szczepan Gajowniczek, Anna Nitkiewicz, Dorota Fopp-Bayat

**Affiliations:** 1Department of Zoology, Faculty of Biology and Biotechnology, University of Warmia and Mazury in Olsztyn, Oczapowskiego 5, 10-718 Olsztyn, Poland; sara.felinska@uwm.edu.pl (S.D.); szczepan.gajowniczek@uwm.edu.pl (S.G.); 2Department of Anatomy, School of Medicine, Collegium Medicum, Warszawska 30 St., 10-082 Olsztyn, Poland; 3Department of Pond Fishery, National Inland Fisheries Research Institute, Oczapowskiego 10, 10-719 Olsztyn, Poland; a.nitkiewicz@infish.com.pl; 4Department of Ichthyology and Aquaculture, Faculty of Animal Bioengineering, University of Warmia and Mazury in Olsztyn, Oczapowskiego 5, 10-719 Olsztyn, Poland

**Keywords:** sterlet, *Acipenser ruthenus*, Toll-like receptor, *TLR2*, *TLR13*, gene expression, polyploids

## Abstract

Toll-like receptors (TLRs) are key components of the innate immune system in fish, responsible for recognizing pathogen-associated molecular patterns derived from bacteria, viruses, and fungi. The sterlet (*Acipenser ruthenus*), an endangered sturgeon species valued for its meat and caviar, is a promising model for studying the effects of polyploidy on immune gene regulation. This study examined the expression of Toll-like receptor type 2 (*TLR2*) and type 13 (*TLR13*) in the heart, liver, gills, spleen, and kidney of diploid and triploid healthy sterlets using real-time PCR. *TLR2* and *TLR13* were expressed in all tissues of both diploids and triploids. In diploids, *TLR2* expression was the highest in the kidney and the lowest in the liver (*p* < 0.05). Similarly, *TLR13* expression in diploids was highest in the kidney and gills, and lowest in the liver (*p* < 0.05). In triploids, no significant tissue-specific variation in *TLR* expression was observed (*p* > 0.05). Comparisons between diploid and triploid sterlets revealed higher *TLR2* expression in the kidney and higher *TLR13* expression in the heart and kidney of diploids (*p* < 0.05). These molecular findings were supported by leukocyte analysis, which showed a significantly lower percentage of lymphocytes and a higher proportion of neutrophils in triploids compared to diploids. Additionally, the proportion of thrombocytes was significantly elevated in triploids (*p* < 0.05). This study provides the first report of *TLR* expression in polyploid fish, offering new insights into immune modulation associated with polyploidy in sturgeons.

## 1. Introduction

The immune system of fish, like that of other vertebrates, comprises both innate and adaptive components that function cooperatively to detect and eliminate pathogens. The innate immune system represents the first line of defense and includes physical barriers, phagocytic cells (e.g., macrophages, neutrophils), antimicrobial peptides, complement proteins, and cytokines such as interleukins and tumor necrosis factor-alpha (TNF-α) [1,2]. Adaptive immunity in fish, although less complex than in mammals, involves T and B lymphocytes, immunoglobulin production, and antigen presentation by specialized cells [2,3]. While adaptive responses are antigen-specific but slower to activate, the innate immune system—particularly through pattern recognition receptors (PRRs)—plays a critical role in rapid pathogen detection and response.

Among PRRs, Toll-like receptors (TLRs) are essential for recognizing pathogen-associated molecular patterns (PAMPs), such as lipopolysaccharides (LPS), flagellin, and viral RNA. Upon detection of PAMPs, TLRs activate signaling cascades that lead to the production of pro-inflammatory cytokines, chemokines, and type I interferons, facilitating both innate and adaptive immune responses [4]. In mammals, 10 to 12 TLRs have been identified [5], whereas fish possess over 20 distinct TLRs, classified into six main superfamilies: TLR1, TLR3, TLR4, TLR5, TLR7, and TLR11 [6,7]. This structural and functional diversity arises from genome/gene duplication events and adaptations to specific environmental pressures. The functional variety of TLRs in fish reflects their evolutionary adaptation to aquatic environments, where they encounter pathogen profiles markedly different from those affecting terrestrial animals [8].

TLRs in fish allow teleosts to recognize a wide range of microbial signals. Among them, Toll-like receptor type 2 (TLR2), part of the TLR1 superfamily, is particularly important due to its ability to form functional heterodimers at the cell surface with other TLRs, such as TLR1, TLR4, TLR6, and TLR 10 [9,10,11]. This ability to heterodimerize not only broadens the range of PAMPs that TLR2 can recognize but also diversifies the downstream signaling pathways [12,13], ultimately enhancing innate immune responses, fostering adaptive immunity, and helping to protect against immune complications following pathogen exposure [14]. TLR2 is known to detect a wide range of PAMPs from viruses, bacteria, fungi, and parasites as well as danger-associated molecular patterns (DAMPs) from damaged host cells [14]. Its downstream signaling, primarily through the MyD88-dependent pathway, results in the induction of pro-inflammatory cytokines and subsequent activation of adaptive immunity. While *TLR2* has been studied in various teleost species [15,16,17,18,19,20,21,22,23,24,25], research on its role in sturgeons remains limited, with only *Acipenser dabryanus* investigated to date [26].

Toll-like receptor type 13 (TLR13) is a member of the TLR11 superfamily and plays a critical role in recognizing both bacterial and viral pathogens. It has been shown to detect a highly conserved sequence of bacterial 23S ribosomal RNA [27] and has been implicated in immune responses to vesicular stomatitis virus infections [28]. Like TLR2, TLR13 also signals via the MyD88-dependent pathway, contributing to pro-inflammatory cytokine production (IL-1β, TNF-α, and IL-10), which is critical for alerting and recruiting other immune cells [29]. While increasing research has focused on *TLR13* expression in fish [30,31,32,33,34,35,36,37,38], only a single study examined its expression in sturgeons (*Acipenser dabryanus*; [26]).

Cytokines, secreted after TLR activation, regulate the immune system by influencing the differentiation, activation, and inflammatory responses of immune cells. Fish possess a wide range of cytokine families, and their gene expression is tightly regulated in response to immune stimuli [39,40]. As summarized by Sakai et al. [40], multiple studies have demonstrated increased expression of cytokine genes in fish treated with bacterial components (e.g., lipopolysaccharide) or immunostimulants such as β-glucans, suggesting their pivotal role in early-stage immune responses and inflammation control. Together, TLRs and cytokines form an integrated molecular network that detects pathogens, initiates inflammation, and orchestrates both innate and adaptive immune responses, ensuring efficient pathogen clearance in teleost fish [41]. This functional responsiveness further highlights the importance of monitoring the expression of these immune mediators to understand the molecular basis of fish immune responses under various physiological conditions, including altered ploidy.

Sturgeons (Acipenseridae) are the most valuable fish species, primarily due to their meat and caviar production [42]. Overexploitation of these products has caused a dramatic decline in natural sturgeon populations, with most species now classified as endangered or near extinction [43]. In addition, bacterial and viral diseases pose serious threats to sturgeon species, particularly in aquaculture settings. Bacterial infections from *Aeromonas* and *Flavobacterium* can cause skin lesions, respiratory dysfunctions, and internal inflammation, while herpesviruses have been associated with neurological disorders [44].

The sterlet (*Acipenser ruthenus*), a small sturgeon species native to European and Asian waters, is an ideal model organism for studying polyploidy, the condition of having more than two sets of chromosomes. Polyploidization, especially triploidization, is of particular interest in aquaculture, as triploid fish (especially females) often exhibit faster growth rates and enhanced meat quality due to their sterility [43]. However, polyploidy increases nuclear DNA content, which may affect gene expression, RNA/DNA ratios, cell size, and immune function [45]. The enlargement of cells and nuclei resulting from additional chromosome sets is offset by a decrease in the total number of cells, including immunological cells [46].

A key aspect of fish immunity involves white blood cells (WBC), including granulocytes (neutrophils, eosinophils, basophils), monocytes, and lymphocytes, which participate in both innate and adaptive immune responses. In addition to these leukocytes, fish thrombocytes—historically regarded primarily as mediators of hemostasis—are now recognized as active participants in innate immunity [47]. Unlike anucleate mammalian platelets, fish thrombocytes are nucleated and have been shown to exhibit phagocytic activity, produce cytokines, and express *TLRs*, such as *TLR5* and *TLR8* [47]. Leukocyte parameters in sturgeons with different ploidy levels have been examined in previous studies [48,49,50]; however, these investigations did not explore their relationship with *TLR* expression. Furthermore, to date, no published studies have reported thrombocyte data comparing diploid and triploid sturgeons.

Despite growing interest in fish immunogenetics, *TLR* expression in sturgeons remains insufficiently characterized, particularly in relation to different ploidy levels. This study aimed to compare the expression of *TLR2* and *TLR13* across immune-relevant tissues (heart, liver, gills, spleen, and kidney) in healthy diploid and triploid sterlets using real-time PCR. Additionally, leukocyte and thrombocyte profiles were examined to evaluate potential changes in circulating immune cells. This is the first report describing *TLR* expression and thrombocyte composition in a polyploid sturgeon species, providing new insights into the immune regulation associated with altered ploidy in fish.

## 2. Results

In the present study, the presence of *TLR2* and *TLR13* transcripts was detected in all examined tissues (heart, liver, gills, spleen, and kidney) of both diploid and triploid individuals of the sterlet *A. ruthenus* (Figure 1). In diploid sterlets, the expression of both *TLRs* varied significantly across tissues. The highest *TLR2* expression was observed in the kidney, while the liver exhibited the lowest expression levels (*p* < 0.05). Similarly, the highest *TLR13* expression in diploids was detected in the kidney and gills, with significantly the lowest expression in the liver (*p* < 0.05). In contrast, no significant differences in the expression of either *TLR* were found across tissues in triploid sterlets (*p* > 0.05).

When comparing *TLR2* expression between diploid and triploid sterlets, significant differences were observed in the kidney, with diploid sterlets showing higher expression levels (Figure 1; *p* < 0.05). For *TLR13*, significant differences in expression were found in the heart and kidney, with diploid sterlets again exhibiting higher mRNA abundance (*p* < 0.05). Additionally, a trend toward higher *TLR13* expression was noted in the gills (*p* = 0.09) and spleen (*p* = 0.08) of diploid fish compared to triploid individuals.

Hematological parameters assessed through blood smear analysis are presented in Table 1. The overall percentage of WBC was significantly lower in triploid sterlets compared to diploids (*p* < 0.001). A marked reduction in lymphocyte proportion was also observed in triploids relative to diploids (*p* < 0.001). Neutrophil granulocyte percentages were significantly higher in triploids than in diploids (*p* < 0.0001), as were thrombocyte counts (*p* < 0.05). No significant differences were found in monocyte or eosinophil percentages between the groups (*p* > 0.05).

## 3. Discussion

This study demonstrates the presence of *TLR2* and *TLR13* transcripts across all examined tissues (heart, liver, gills, spleen, and kidney) in both diploid and triploid sterlets (*A. ruthenus*). These findings are consistent with prior research, showing that *TLRs* are broadly expressed in various fish species and tissues. The presence of *TLR2* was previously detected in common carp (*Cyprinus carpio* [22]), grass carp (*Ctenopharyngodon idella* [23]), rohu (*Labeo rohita* [19]), channel catfish (*Ictalurus punctatus* [17]), orange-spotted grouper (*Epinephelus coioides*, [18]), Japanese flounder (*Paralichthys olivaceus* [15,25], mandarin fish (*Siniperca chuatsi* [34]), and Dabry’s sturgeon (*Acipenser dabryanus* [26]). Similarly, *TLR13* has also been reported in several fish, including miiuy croaker (*Miichthys miiuy* [30]), mandarin fish (*Siniperca chuatsi* [34]), golden pompano (*Trachinotus ovatus* [37]), large yellow croaker (*Larimichthys crocea* [33]), orange-spotted grouper (*Epinephelus coioides* [31]), *Nibea albiflora* [38], and soiny mullet (*Liza haematocheila* [32]). The widespread presence of these receptors across species and tissue types underscores their conserved role in pathogen detection and immune signaling in fish.

In diploid sterlets, *TLR2* expression was the highest in the kidney, a result that aligns with previous studies emphasizing the importance of the kidney as a key immune organ in fish, involved in hematopoiesis and immune surveillance [51]. Similar high mRNA expression levels of this receptor have been reported in the kidney of Dabry’s sturgeon [26], turbot [21], common carp [22], grass carp [23], Japanese flounder [15], and mandarin fish [34]. The results from the present as well as the previous studies suggest that the kidney functions as a critical site for immune signaling due to its role in filtering pathogens from the bloodstream [51]. By contrast, the lowest *TLR2* expression observed in the liver could be attributed to the liver’s primary metabolic function, which may limit its involvement in frontline immune defense compared to other organs like the kidney or gills. A similar pattern of low *TLR2* expression in the liver has been reported in grass carp [23], while Baoprasertkul et al. [17] observed relatively high *TLR2* expression in the liver of healthy catfish compared to other tissues. These differences may be due to methodological variations; Baoprasertkul et al. [17] used conventional PCR, a method with lower resolution than real-time PCR.

The expression pattern of *TLR13* was highest in the kidney and gills in diploid sterlets. Similar findings were reported by Tang et al. [26], who observed high expression of this receptor in the kidney, gills, and heart of Dabry’s sturgeon; however, there is no information on whether these differences were statistically significant. High levels of *TLR13* mRNA have also been found in the kidneys of several other fish species, including miiuy croaker [30], golden pompano [37], large yellow croaker [33], orange-spotted grouper [31], and soiny mullet [32] as well as in the gills of the orange-spotted grouper [31]. In contrast, low levels of *TLR13* expression were observed in the gills of miiny croaker [30] and soiny mullet [32].

The high expression of *TLR13* in the gills aligns with the role of gills as a primary site of pathogen entry in aquatic organisms, where they serve as the first line of defense against pathogens. TLRs in the gills likely play a crucial role in detecting waterborne pathogens, making the gills an essential component of the fish’s immune system [6]. The high expression of *TLR13* in the kidney further supports its involvement in systemic immune responses, especially in detecting bacterial RNA, as *TLR13* is known to recognize conserved sequences in bacterial 23S rRNA [27,35]. Lower expression of *TLR13* in the liver parallels the patterns observed in this study for *TLR2* and is consistent with previous findings in the orange-spotted-grouper [31] and soiny mullet [32]. On the other hand, higher levels of *TLR13* expression have been reported in the liver of large yellow croaker [33], *Nibea albiflora* [38], and miiuy croaker [30]. The variability in *TLR13* expression patterns observed across studies suggests that *TLR13* expression may be species-specific.

Interestingly, no significant differences in *TLR2* or *TLR13* expression were observed across tissues in triploid sterlets. To our knowledge, this is the first report about *TLR* expression in polyploid fishes. This lack of tissue-specific variation in triploids might be due to physiological and genomic changes associated with polyploidy. It has been demonstrated that triploid induction in *A. altiparanae* specimens resulted in immune impairments and potentially lower resistances to disease and low-quality environments [52]. Moreover, Cadonic et al. [53] provide evidence for epigenetic dysregulation in triploid fishes, which may contribute to their poor performance in response to stress. Thus, the flattened *TLR* expression profile observed here may reflect diminished immunological plasticity in triploid sterlets.

When comparing gene expressions between diploid and triploid sterlets, *TLR2* levels were similar in the heart, liver, gill, and spleen, but significantly higher in the kidney of diploids. Higher *TLR2* expression in diploids suggests that polyploidy may influence the immune response capacity of specific organs, with diploids possibly having a more robust immune response in this certain tissue. Christensen et al. [45] found that many genes in the liver exhibit similar expression levels between diploid and triploid coho salmon, likely due to a balance in mRNA transcript production per gene copy (positive gene dosage effects), even in the larger cells of triploids. On the other hand, several genes were differentially expressed between diploid and triploid salmon, indicating that some loci are sensitive to cell size and/or DNA content per cell [45].

For *TLR13*, diploid sterlets exhibited significantly higher expression in the heart and kidney, with a trend toward higher expression in the gills and spleen in comparison to triploids. The higher expression of *TLR13* in diploids may reflect a greater readiness of their immune system to recognize and respond to RNA-based pathogens, which could confer a selective advantage in environments with high pathogen loads [28]. However, more studies are needed to confirm this hypothesis.

Our results showed a significantly lower percentage of WBC and a decreased proportion of lymphocytes as well as a higher percentage of neutrophils and thrombocytes in triploids compared to diploid sterlets. These findings are consistent with previous studies on sturgeon species. Wlasow and Fopp-Bayat [48] as well as Rożyński et al. [49] reported differences in leukocyte parameters between diploid and triploid Siberian sturgeons (*A. baerii*). Additionally, Salkova et al. [50] observed altered immune cell profiles across species with different ploidy levels, including diploid *A. ruthenus*, tetraploid *A. gueldenstaedtii*, and hexaploidy *A. brevirostrum*. Moreover, these scientists suggest a significant effect of ploidy level on the total number of leukocytes and morphological nuclear changes in the granulocytes and lymphocytes in sturgeon species [50]. To our knowledge, this is the first study to report thrombocyte profiles in triploid sturgeons. Similarly, Gao et al. [54] observed an increased percentage of thrombocytes in triploid and tetraploid loaches (*Misgurnus anguillicaudatus*); however, the statistical significance of these findings was not provided. This higher proportion of thrombocytes in triploid sterlets may indicate possible differences in hematopoietic regulation. Fish thrombocytes are fascinating because they blur the line between blood clotting and immune function, much more so than in mammals [47].

The altered leukocyte profiles observed in the present study in triploid sterlets might support the hypothesis that polyploidy influences immune system function. Blood smear analysis revealed a significantly lower percentage of WBC in triploid individuals compared to diploids, along with notable shifts in leukocyte and thrombocyte composition. Most strikingly, triploids exhibited a marked reduction in lymphocyte percentage, which may indicate compromised adaptive immune potential, as lymphocytes are essential for antigen-specific responses [39]. At the same time, triploids showed elevated proportions of neutrophils and thrombocytes—cells primarily associated with early-phase, non-specific immune responses [39,47]. These cellular changes, combined with the reduced and less tissue-specific expression of *TLR2* and *TLR13* observed in triploid sterlets, suggest that polyploidy may result in a less efficient or less tightly regulated immune response. While the increased abundance of thrombocytes and neutrophils in triploids may reflect a compensatory mechanism for lower lymphocyte levels, it remains unclear whether such a response is sufficient to maintain effective immune function.

One important limitation of this study is the lack of histological or morphometric data on cell size and number. Such data could further clarify whether the observed gene expression patterns are associated with structural changes in immune organs. Although we included blood smear analysis, more detailed histological analyses will be needed to evaluate tissue-level adaptations to polyploidy. Furthermore, the study was limited to the assessment of two *TLR*s. While TLR2 and TLR13 are important components of innate immunity, additional immune markers, such as cytokines, would provide a more comprehensive understanding of the immune landscape in polyploid fish. Further studies using transcriptomics approaches, cytokine assays, and infection models are needed to clarify the functional significance of the observed gene expression patterns.

Despite these limitations, this study presents the first analysis of *TLR* gene expression alongside immune cell composition in diploid and triploid sterlets. These findings establish important groundwork for future research on the functional immunology of polyploid sturgeons and highlight the need to evaluate their immune competence, particularly in aquaculture settings where disease resistance is critical.

## 4. Materials and Methods

This study was conducted in strict accordance with the Polish ACT of 21 January 2005 on Animal Experiments (Dz. U. of. 2005, No 33, item 289) and was approved by the Local Ethical Committee for the Experiments on Animals at the University of Warmia and Mazury in Olsztyn, Poland (Permit Number: 75/2012).

### 4.1. Fish and Tissue Collection

The diploid (2n) and triploid (3n) sterlets were produced according to the procedure published by Fopp-Bayat et al. [55], using sterlet eggs collected from two sterlet females (Female I and Female II) aged 6+ years and fresh sperm collected from sterlet male aged 6+ years. Fish reproduction was conducted under controlled conditions at the Wasosze Fish Farm near Konin, Poland. The portion of 4000 eggs collected from two females were mixed before fertilization and divided into two experimental groups: diploid (2n) and triploid (3n). Sperm collected from two males were examined under a light microscope, and sperm with more than 90% spermatozoa motility were selected to fertilize sterlet eggs. One portion of fertilized eggs (approx. 2000 eggs) was induced to triploidization [55], while the second portion (2n) was the control group. Fertilized eggs from the 2n and 3n groups were incubated in cage incubators [56] at a constant temperature of 17 °C until hatching. The hatched diploid (2n) and triploid (3n) larvae were transferred to two 40 L tanks in a recirculating aquaculture system (RAS) at the Centre for Aquaculture and Environmental Engineering of the University of Warmia and Mazury in Olsztyn. The experimental fish were reared for twelve months post-hatching (mph) according to the procedure described by Fopp-Bayat et al. [57] until sampling.

A total of 11 healthy sterlets *Acipenser ruthenus* (6 diploids and 5 triploids; F1 generation) with an average body weight of 401 g and 408 g, respectively, were used in the experiment conducted from 2020 to 2021. The ploidy level of each fish was confirmed using fin-clips and flow cytometry. Based on the ploidy analysis, the fish were divided into two experimental groups. Blood for hematological analysis was collected via caudal vein puncture using a heparinized syringe after using MS222 (100 mg/L). After dissection, tissue samples (heart, liver, gill, spleen, and kidney) were immediately collected, frozen in liquid nitrogen, and stored at −80 °C until RNA isolation. An overview of the experimental design is presented in Figure 2.

### 4.2. Ploidy Verification

The ploidy status of the fish was determined using a CyFlow Ploidy Analyzer (Sysmex) and a ready-to-use kit for nuclei extraction and nuclear DNA staining (CyStain UV Precise T, Sysmex Partec GmbH, Görlitz, Germany). Some modifications were made to the standard protocol. A small piece of dorsal fin (~2 mm) was collected from each fish and placed in a small Petri dish containing 0.3 mL of nuclear extraction buffer. The tissue was mechanically minced using two scalpels. The resulting suspension was transferred to 1.5 mL Eppendorf tubes and incubated at room temperature for 5 min with occasional mixing (by finger vortex). The suspension was then filtered through a 30 µm nylon filter (CellTrics, Sysmex Partec GmbH, Görlitz, Germany) into cytometric tubes. After filtration, 700 µL of staining buffer containing 4′,6-diamidino-2-phenylindole (DAPI) was added to the isolated nuclei, which were incubated for 3 min in reduced light conditions. The samples were then analyzed. Ploidy status was determined by comparing relative DNA content with a standard haploid DNA content (1C) obtained from the sperm of *A. ruthenus*.

### 4.3. Total RNA Extraction and cDNA Synthesis

Total RNA was extracted from each tissue using the Total RNA Mini Kit (A&A Biotechnology, Gdynia, Poland) following the manufacturer’s protocol, with slight modifications. Tissue samples weighing 10–20 mg were homogenized in 2 mL tubes containing 800 µL of Phenozol and ceramic beads (Blirt S.A., Gdansk, Poland). Homogenization was performed using a MagNALyser tissue homogenizer (Roche Diagnostics GmbH, Mannheim, Germany). RNA was eluted with 50 µL of RNase- and DNase-free water at 50 °C. RNA integrity was confirmed by agarose gel electrophoresis, and RNA quality and concentrations were assessed spectrophotometrically (Nanodrop, Thermo Fisher Scientific Inc., Waltham, MA, USA) by measuring absorbance at 260 nm and 280 nm. RNA samples were stored at −80 °C until further use. The cDNA template was synthesized from approximately 1 µg of total RNA using the QuantiTect Reverse Transcription Kit (Qiagen GmbH, Hilden, Germany) in accordance with the manufacturer’s instructions. The protocol included a genomic DNA removal step using genomic DNA wipe-out buffer provided in the kit. As a control for genomic DNA contamination, reactions without Quantiscript Reverse Transcriptase were performed for each sample. Reverse transcription was carried out for 30 min at 42 °C, followed by a termination step at 95 °C for 3 min (Veriti, Applied Biosystems, Thermo Fisher Scientific, Waltham, MA, USA). The resulting cDNA was diluted 1:10 before being used in real-time PCR analysis.

### 4.4. Real-Time PCR (qPCR)

Real-time PCR was performed in 96-well plates using a QuantStudio 5 Real-Time PCR System (Thermo Fisher Scientific, Waltham, MA, USA) with Power SYBR Green PCR Master Mix (Life Technologies, Carlsbad, CA, USA). Two reference genes, *EF1α* and *RPL13*, were used for normalization [58]. To evaluate reference gene expression stability, the geNORM software tool (version 3.2, Biogazelle, Zwijnaarde, Belgium) was used. Specific primers for *TLR2* and *TLR13* were designed using PRIMER3 based on the available mRNA sequences of both sterlet receptors from the NCBI database (accession numbers: XM_034014252.2, XM_034908419.1, respectively). Primer details are provided in Table 2. Primer specificity was confirmed via melting curve analysis and verification of amplicon size by gel electrophoresis. For each assay, primer efficiency was determined by a standard curve of cDNA samples according to the MIQE guidelines for qPCR [59]. The linear correlation coefficient (*R*^2^) ranged from 0.985 to 0.998, and the PCR efficiency varied from 99.3% to 116.8%. Non-template control (NTC) samples were included in each run to monitor for contamination. To ensure the absence of genomic DNA amplification, control reactions without reverse transcriptase were performed. Relative mRNA expression levels were calculated using the comparative cycle threshold (CT) method [60].

### 4.5. Blood Smears

Blood smears were prepared immediately after collection, air-dried at room temperature, fixed in methanol, and stained using a procedure published by Svobodová et al. [61]. Slides were observed under a light microscope (Eclipse 80i, Nikon, Tokyo, Japan) equipped with NIS-Elements software (version 4.6, Nikon, Tokyo, Japan) using a digital camera (DS-Fi1c, Nikon, Tokyo, Japan). For each specimen, 200 nucleated immune cells were counted and identified based on cell size, nuclear morphology, and cytoplasmic staining. Leukocyte and thrombocyte differentiation was expressed as a percentage of each cell type.

### 4.6. Statistical Analysis

Statistical analysis was performed using the Statistica software (version 13.3, StatSoft Inc., Tulsa, OK, USA). A one-way ANOVA was used to compare the relative expression of *TLR2* and *TLR13* between tissues within diploid and triploid sterlets, followed by a least significant difference (LSD) test. To compare the relative expression of *TLR2* and *TLR13* between diploid and triploid individuals within each tissue, an independent samples *t*-test was employed. Leukocyte and thrombocyte count data were also analyzed using Student’s *t*-test. Data were log-transformed when the variances of the compared means differed by at least one order of magnitude. Differences were considered statistically significant at *p* < 0.05.

## Figures and Tables

**Figure 1 ijms-26-03986-f001:**
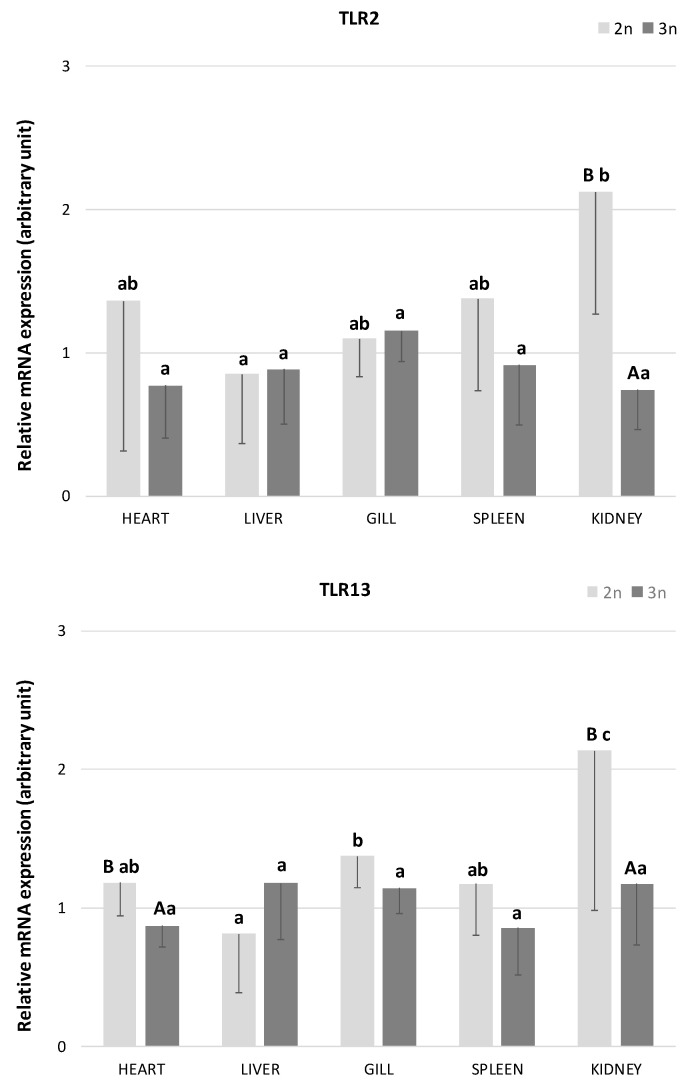
The relative mRNA expression of Toll-like receptor 2 (*TLR2*) and 13 (*TLR13*) in heart, liver, gill, spleen, and kidney in diploid (*n* = 6 adult fish) and triploid (n = 5 adult fish) sterlet *Acipenser ruthenus*. Bars indicate average gene expression of fishes at each ploidy level, and error bars indicate standard deviations. Different small letters indicate statistically significant differences between tested tissues within diploid or triploid group (*p* < 0.05, ANOVA); different capital letters indicate statistically significant differences between diploid and triploid groups within the examined tissue (*p* < 0.05, independent-samples *t*-test).

**Figure 2 ijms-26-03986-f002:**
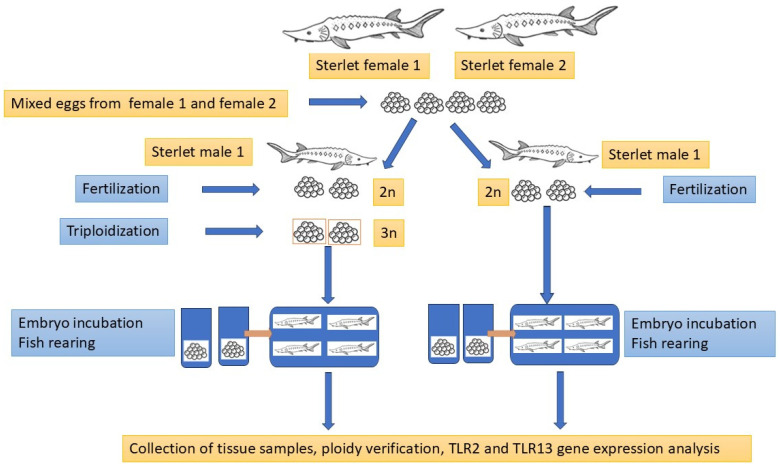
The graphical illustration of the experimental design.

**Table 1 ijms-26-03986-t001:** Comparison of hematological parameters between diploid and triploid sterlets (*Acipenser ruthenus*).

Immune Nucleated Cells Percentage (%)	Diploid Sturgeon	Triploid Sturgeon	*p*
WBC	7.78 ± 0.42	4.73 ± 0.66	<0.001
Lymphocytes	46.13 ± 2.55	20.78 ± 2.57	<0.001
Monocytes	3.33 ± 0.55	4.83 ± 1.02	N
Neutrophils	17.35 ± 0.23	26.23 ± 0.94	<0.0001
Eosinophils	2.5 ± 0.46	3.15 ± 0.3	N
Thrombocytes	30.68 ± 1.69	45.10 ± 4.33	<0.05

Data presented as average value ± SD (standard deviation); WBC: white blood cells, *p*: statistical significance, N: not significantly different at *p* > 0.05.

**Table 2 ijms-26-03986-t002:** Primers used for real-time PCR.

Gene Name	Primer Sequences (F: Sense, R: Antisense)	Amplicon Length (bp)	Concentration of Primers (nM): Sense/Antisense
*TLR2*	F: CTCTCGGAGCACTTTGTTCG R: ACTGCCCTCTGTCCTTCATC	212	200/200
*TLR13*	F: ATACAACACGCACGATGAGC R: TAGTGGTGGCTGATGATGCA	180	400/200
*EF1α*	F:GGACTCCACTGAGCCACCT R: GGGTTGTAGCCGATCTTCTTG	89	200/400
*RPL13*	F:GGACGTGGTTTCACCCTTG R: GGGAAGAGGATGAGTTTGGA	166	200/400

## Data Availability

The raw data supporting the conclusions of this article will be made available by the authors upon request.

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
