# Peer review of "Toll-like Receptor Type 2 and 13 Gene Expression and Immune Cell Profiles in Diploid and Triploid Sterlets (Acipenser ruthenus): Insights into Immune Competence in Polyploid Fish"

_ijms, 2025, doi:10.3390/ijms26093986_

Round 1
Reviewer 1 Report (Previous Reviewer 2)
Comments and Suggestions for Authors
This manuscript barely makes a coherent whole but may be accepted as a Short Communication!
Author Response
Reviewer 1
Comments and Suggestions for Authors
This manuscript barely makes a coherent whole but may be accepted as a Short Communication!
Response:
Thank you for your comment. We agree that the scope and format of this paper are most appropriate for a Short (or Bief) Communication.

Reviewer 2 Report (Previous Reviewer 1)
Comments and Suggestions for Authors
"The study provides valuable insights into TLR expression and immune function in diploid and triploid sterlets. While the work is comprehensive, it could benefit from minor revisions, such as expanding the introduction to include a broader range of immune-related genes and incorporating additional immune markers like cytokines. Given the nature of the study, histological data on cell size and number would have further enhanced the understanding of how polyploidy influences immune system function. I would also recommend that in the discussuin, the authors explicitly state any limitations of the study, particularly regarding the scope of the immune markers examined and the absence of histological data, to provide a clearer context for the findings.
Author Response
Reviewer 2
Comments and Suggestions for Authors
"The study provides valuable insights into TLR expression and immune function in diploid and triploid sterlets. While the work is comprehensive, it could benefit from minor revisions, such as expanding the introduction to include a broader range of immune-related genes and incorporating additional immune markers like cytokines. Given the nature of the study, histological data on cell size and number would have further enhanced the understanding of how polyploidy influences the immune system function. I would also recommend that in the discussuin, the authors explicitly state any limitations of the study, particularly regarding the scope of the immune markers examined and the absence of histological data, to provide a clearer context for the findings.
Response:
Thank you for this thoughtful and constructive feedback. In response, we have expanded the Introduction section to include additional context on key immune-related markers, with a particular focus on cytokines, which play a crucial role in the immune responses following TLR activation. As suggested, we have also revised the Discussion section to address the study’s limitations, particularly the limited scope of immune markers studied and the lack of available histological data. In the current manuscript, the additional valuable information was included in the Introduction section – Lines: 76-77 and 80-91; and in the Discussion section – Lines: 270-283. Additionally, we supplemented the References List with three additional references (numbers: 39, 40, 41 in the updated manuscript) to support these revisions.

This manuscript is a resubmission of an earlier submission. The following is a list of the peer review reports and author responses from that submission.
Round 1
Reviewer 1 Report
Comments and Suggestions for Authors
This work is interesting and presents some novel findings, but it has several weaknesses and gives the impression of being incomplete. The focus on TLR receptors, while justified, would benefit from a more comprehensive introduction to genes related to fish immunity. This introduction would provide necessary context before narrowing the focus to TLRs. Moreover, since the functional significance of changes in TLR gene expression remains limited, it is difficult to fully justify such an experimental design without exploring broader immune-related factors.
While TLRs play a critical role in recognizing pathogens and triggering immune responses, they represent only one component of the immune system. To gain a more complete understanding of the fish's immune health, the study should incorporate additional immune markers, such as cytokine profiles (e.g., IL-1β, TNF-α), immune cell counts (e.g., macrophages or neutrophils), and complement system activity. Without examining a broader range of immune indices, the study may overlook how polyploidy impacts the functional immune capacity of the fish, including their ability to respond to infections or environmental stressors. It is crucial to test whether the observed differences in TLR expression lead to actual changes in immune function, such as cytokine production or pathogen resistance.
Polyploidy results in increased DNA content, which can affect gene expression, RNA/DNA ratios, cell size, and cell number. In polyploid organisms, cells often have larger volumes due to the increased DNA content. This may influence RNA abundance and, consequently, gene expression. However, without histological data to assess cell number and size, it is difficult to fully interpret the impact of polyploidy on gene expression.
The findings related to immune responses in triploid sterlets are valuable for aquaculture, but it would be beneficial to emphasize the need for further studies to evaluate immune functionality in triploid fish. While polyploidy is often associated with sterility, the broader effects on immune system capacity remain underexplored and may be a significant concern for aquaculture practices. Additional data on disease susceptibility or stress resilience in triploid fish would enhance the impact of this research.
Reviewer 2 Report
Comments and Suggestions for Authors
Even though this is a short report the results displayed in this manuscript is of extremely minor interest for readers of a journal. Two Tolls mRNA are investigated in some tissues and there is hardly any significant differences in their expression or at least not any interesting difference. This minmal study should be incorporated in a larger study and not publishes as it is.
Comments on the Quality of English LanguageWhy is this necessary to include comments on the English which already has a box to fill in above?